# From-Toilet-to-Freezer: A Review on Requirements for an Automatic Protocol to Collect and Store Human Fecal Samples for Research Purposes

**DOI:** 10.3390/biomedicines11102658

**Published:** 2023-09-28

**Authors:** Frances Widjaja, Ivonne M. C. M. Rietjens

**Affiliations:** Division of Toxicology, Wageningen University & Research, 6708 WE Wageningen, The Netherlands; ivonne.rietjens@wur.nl

**Keywords:** human, gut microbiota, in vitro, protocol, fecal samples, collection, storage

## Abstract

The composition, viability and metabolic functionality of intestinal microbiota play an important role in human health and disease. Studies on intestinal microbiota are often based on fecal samples, because these can be sampled in a non-invasive way, although procedures for sampling, processing and storage vary. This review presents factors to consider when developing an automated protocol for sampling, processing and storing fecal samples: donor inclusion criteria, urine–feces separation in smart toilets, homogenization, aliquoting, usage or type of buffer to dissolve and store fecal material, temperature and time for processing and storage and quality control. The lack of standardization and low-throughput of state-of-the-art fecal collection procedures promote a more automated protocol. Based on this review, an automated protocol is proposed. Fecal samples should be collected and immediately processed under anaerobic conditions at either room temperature (RT) for a maximum of 4 h or at 4 °C for no more than 24 h. Upon homogenization, preferably in the absence of added solvent to allow addition of a buffer of choice at a later stage, aliquots obtained should be stored at either −20 °C for up to a few months or −80 °C for a longer period—up to 2 years. Protocols for quality control should characterize microbial composition and viability as well as metabolic functionality.

## 1. Introduction

Intestinal microbiota are important for host health because they play a role in a variety of physiological and metabolic pathways that influence health and disease [1,2,3,4,5]. Appendix A provides an overview of literature studies reporting the relationship between fecal microbiota and health or disease at phylum, family, genus and species levels. This includes studies on the role of gut microbiota in sleep apnea, type 2 diabetes (T2D), inflammatory bowel disease (IBD), irritable bowel syndrome (IBS), anorexia, obesity, colorectal cancer (CRC), autism, Parkinson’s disease and hepatic encephalopathy. For instance, the genus *Roseburia* sp. has been correlated with the health benefits of fiber consumption due to its ability, together with other commensal bacteria, to produce short-chain fatty acids (SCFAs) [6]. In reverse, the absence or decline of the same bacteria could indicate disease symptoms such as pouchitis for patients with ulcerative colitis (UC) [7]. Another example includes *Parvimonas micra*, *Fusobacterium nucleatum* and *Lachnoclostridium* sp. that are commonly used in detection of colorectal cancer (CRC) [8].

In addition to composition, the viability and metabolic functionality of the fecal microbiota, including the chemical reactions that these microbes are able to perform or facilitate, also play a role in health and disease [9,10,11,12]. Appendix A provides an overview of chemical conversions catalyzed by the microbiota. Gut microbiota, for example, play an important role in bile acid metabolism and homeostasis, while changes in bile acid profiles and size of the bile acid pool have been reported to show associations with diseases such as obesity, bile acid diarrhea (BAD), IBS, IBD, T2D, *C. difficile* infection (CDI) and hepatocellular carcinoma [13,14,15,16].

Studies on intestinal microbiota are often based on fecal samples because (i) microbiota present in fecal samples closely resemble those in colonic samples [17], where >70% of the intestinal bacteria reside [18,19], and (ii) human feces can be easily obtained in a non-invasive manner [20]. This implies that studying fecal samples to obtain insight into the role of the intestinal microbiota in health and disease requires adequate sampling, processing and storage. In order to facilitate high-throughput collection and storage, an automatic protocol to collect and store human fecal samples for research purposes would be recommended. The current mini-review presents an overview of data currently available on requirements to consider when developing such a protocol. Publications were systematically reviewed for different topics to be considered when collecting and storing human fecal samples for research purposes, starting from the fecal material donor, the toilet itself, the conditions required for fecal collection, processing, storage and, lastly, the quality control.

## 2. Differences Amongst Reported Human Fecal Microbiota Protocols

Despite the importance and relevance of human fecal microbiota studies, laboratories worldwide perform their fecal collection, processing and storage in a different manner, as previously summarized [21,22,23,24]. While it is customary to change protocols based on individual research needs, from comparison of these protocols, the critical steps to be considered become clear. Available data also suggest what would be the optimal requirements for each of these critical steps in order to define a harmonized automated protocol. These critical steps and their relevant parameters and requirements are discussed in the following sections in sequential order.

### 2.1. Criteria for Human Fecal Material Donors

Before a study begins, a section of the human population is targeted as fecal material donor (i.e., inclusion criteria). These candidates are often invited to fill up a questionnaire on their health status, diet, lifestyle, gastrointestinal issues, antibiotic and medication usage and other factors, which can be used to screen their suitability as fecal donors (i.e., exclusion criteria). This step has been previously documented in detail [25,26] but not all studies describe how to ensure that fecal donors truthfully answer the questionnaire rather than provide answers that allow them to donate their fecal materials [26]. In addition, the categories and the extent of such exclusion criteria differs per research group. For instance, while travel history, gastrointestinal diseases, pregnancy and recent antibiotic usage are commonly included [27,28,29], screening based on respiratory diseases, positive tests for certain bacterial or viral infections and mental health condition is uncommon [25]. Nevertheless, data collected on screened fecal donors would never fully capture the entire extent of an individual’s lifestyle, consequently being a black box in downstream data analysis. In addition, most studies do not disclose their screening criteria, making it difficult to reproduce or compare results. Therefore, required criteria for participants’ information should be written as formal guideline such as those published by The Organization for Economic Co-operation and Development (OECD), the World Health Organization (WHO) or the European Food Safety Authority (EFSA).

In addition to the inclusion/exclusion criteria, the number of recruited fecal donor participants varies, with, for instance, ≤30 individual [30,31,32] or significantly greater than 30 individuals [33,34,35] recruited in previous studies. While a low number of human fecal participants may often be sufficient to reach the study’s objective, and a high number of participants can increase results’ representativeness, recruiting a high number of participants and manually processing the fecal samples are laborious in studies with very high numbers of fecal donors [22,36,37], expressing the need for a higher-throughput processing stream.

### 2.2. Urine–Feces Separation

The first critical factor in a high-throughput automated protocol would be the urine–feces separation. Smart toilets that separate urine from feces already exist, mainly for collecting user’s health information, developing sustainable cities, mineral recollection and usage in emergency camps [38,39,40,41,42,43,44,45]. Some toilets are equipped to meet women’s and men’s different preferences on urine excretion angle, due to different physiology such as height and length of feet [46], and contain sensors that allow self-analysis of human excreta as personalized health monitoring systems [42,44,46]. Different technologies have been used to separate urine and feces such as a vacuum suction pump with two separate holes [47], a bi-sloped conveyor belt to two different tanks [38], a urine diverting pedestal [41], a three-chamber septic tank (Aquatron separation technique (Aquatron, Västerås, Sweden)) [40] and an urine diversion equipped with a sensor such as the Beevi Toilet (Ulsan, Republic of Korea) developed by Science Walden [43]. However, all these technologies are not yet widely applied by research groups performing human gut microbiota studies, implying that most fecal donor participants still have to defecate in a normal toilet without urine–feces separation. While it is possible to defecate without urinating, it is difficult because, once the stronger anal sphincter relaxes, the weaker urinary sphincter also relaxes, which allows urine to pass at the same time as stool excretion [48]. Thus, for efficient urine–feces separation, application of a smart toilet that can automatically separate urine and feces during human fecal excreta collection seems a prerequisite. Nevertheless, in many places where research takes place, such a device may not be a realistic option and one would have to turn to simpler solutions.

### 2.3. Collection of the Fecal Sample

Collection of the fecal sample from the smart toilet should preferably also be performed in an automatic way, eliminating the need for the donor to collect and transfer the sample by him/herself. For example, Zymo feces catcher (Zymo Research, Kempen, Germany), Fisherbrand™ Commode Speciment Collection System (Fisherbrand™, Landsmeer, The Netherlands) and Fe-Col^®^ Faecal Sample Collection Kits (Alpha Laboratories, Eastleigh, Hampshire, UK) still require manual transfer. The question to be answered then is whether to collect the sample as a whole or to store it in aliquots. Given that stools as such are generally large compared to the amount of material needed for subsequent analyses, aliquoting is implied. Aliquoting avoids multiple freeze–thaw cycles that cause DNA degradation [22], and innovations such as CryoXtractR (Eppendorf, Hamburg, Germany) were developed to facilitate this aliquoting [49]. While the effect of one freeze–thawing cycle on stool microbiota composition is not significant, it is important to note that subsequent freeze–thawing cycles may reduce the overall levels of viable bacteria [50]. Therefore, fecal sample processing protocols should divide specimens into smaller aliquots as a precautionary principle (i.e., avoiding DNA degradation and limiting the decrease in viability) and as a pragmatic approach (i.e., resulting in shorter thawing time at room temperature (RT)).

The need to aliquot also brings the question as to the need for homogenization. There are conflicting opinions related to how homogenized or well-mixed collected fecal specimens already are or whether they should be processed to homogenize their content. One study determined the effect of stool structure on microbial composition by comparing the inner and outer layers of the samples, as well as stool water content, and found minor differences in microbial composition and species abundance between both layers and between different water percentages [23]. Yet the use or omission of bead-beating to homogenize stool samples resulted in aliquots with different proportions of Gram-positive and Gram-negative bacteria [23,24,51]. In contrast, large inconsistency of taxa within an individual stool was found, possibly due to spatial variation of microenvironments [52]. Another study argued that stools with hard and lump consistency showed more diverse microbial composition across different collection sites in a single bowel movement [22,52,53,54], and the outer part and inner part of stool contain a different ratio of aerobes/anaerobes due to oxygen concentration gradient [22]. These results suggest that stool homogenization under anaerobic conditions is essential to level out intra-sample variation. Stool homogenization is also crucial for metabolomic (metabolite) analysis but not for microbiome (microbiota) analysis because different sampling regions did not show differences in microbial community alpha diversity, while some identified that metabolites varied significantly across different sampling regions [55]. Considering that homogenization might be crucial for particularly hard and lumpy stool (Type 1–2 according to the Bristol stool chart) [56] but remains an unpleasant task to be manually performed and that metabolomics studies might require fecal homogenization to reduce unnecessary variability, automated homogenizer equipment [57] should be used such as a meat grinder, fecal matter blender [58] or sewage grinder system [59], preferably under anaerobic conditions to avoid preferential loss of obligate anaerobic bacteria.

### 2.4. Anaerobic Conditions

Anaerobic bacteria account for most of the fecal bacteria and are 100–1000 times more numerous in the intestinal microbiota pool than aerobic bacteria [60,61] but may die during sample preparation due to delay between stool collection and transport to anaerobic laboratory conditions [58,62]. Some studies prevent this loss of anaerobes by storing stools in a closed box with direct exposure to an anaerobic system such as Anaerocult^TM^ A (Merck, Darmstadt, Germany) to ensure an oxygen-free environment before transferring the fecal-containing box into an anaerobic chamber [63,64]. Another system called Exakt Pak canister (Inmark Packaging, Austell, GA, USA) was used to collect stool but there was no indication that the system contained an anaerobic environment [65]. One study performed bacterial enumeration with a different anaerobic system called GasPak generator (Becton Dickinson, Sparks, NJ, USA) [58]. They reported that the time between sample collection and sample processing was an important parameter and that this time should be as short as possible because bacterial culturability showed no discordance when samples were exposed to oxygen for less than two minutes; accordingly, the rest of the work was performed in the anaerobic chamber [58]. In addition, bacterial culturability increased when antioxidant solutions were used [58]. A recently patented fecal sampling device enables anaerobic and hygienic self-collection of stool samples for further processing without opening the device, preventing contact with the fecal donor and sample exposure to the environment [66]. All in all, these studies point to the need for an anaerobic environment for collecting fecal bacteria and a quick transport from toilet to the anaerobic laboratory conditions, which may best be achieved by using a vacuum toilet as used in other fields such as commercial aircraft [67]. This technology could automatically transport the feces to a compartment that can be made anaerobic. Alternatively, the patented stool self-collection device could achieve this objective [66], although at present it is not yet commercially available and the collection is not high-throughput.

### 2.5. Buffer or No Buffer in Fecal Material Processing

The usage of buffer or solution used to dissolve and subsequently store fecal material differs per study. A previous study summarized an extensive list of different buffer types and protocols used that appeared to depend on the type of experiments performed such as metabolomics, 16 s rRNA sequencing, metatranscriptomics or quantification of bacterial enzyme activities [21]. In addition, when freezing samples for storage at low temperature, usage of 10–20% (*v*/*v*) glycerol as cryoprotectant was encouraged to preserve bacteria because freeze–thawing compromises cell viability [21,68]. Yet another study advocated against dissolving fecal materials in 10% (*v*/*v*) glycerol for downstream lyophilization because it leads to an insufficiently dry and sticky product [69]. High concentrations of cryoprotectants such as glycerol and dimethylsulfoxide may penetrate cell membranes and compromise original biological and metabolic characteristics of fecal materials, leading to alternative cryoprotectants such as maltodextrin and trehalose [69], although the effect of cryoprotectant penetration on bacterial viability remains to be elucidated [68]. In addition, although glycerol might protect cell viability upon freezing [68], upon thawing, it may cause medium enrichment, leading to an increase in certain bacteria and, consequently, changing the original composition or viability of microbiota [70].

In addition to cryprotectants, the inclusion of stabilizers such as RNAlater (Sigma-Aldrich, St. Louis, MO, USA) is debatable. On the one hand, RNAlater is known to protect stool RNA and DNA from rapid degradation, even after up to five years of storage [22,71]. However, the usage of RNAlater was negatively recommended as a storage medium of stool samples for microbial and metabolomic analysis due to a significant level of variation [55]. While RNAlater or other non-lytic or lytic buffers may protect microbial DNA, usage of such buffers may greatly decrease the viability of living cells and hamper subsequent culturing and colonizing studies [72,73].

All in all, rapid deep-freezing to −80 °C without buffer was strongly suggested as the best practice [70,74] because there was no alteration in microbial community with respect to composition between samples from direct freezing without buffer at −80 °C, 24 h storage at −20 °C and 24 h storage at 4 °C or RT [75] and no alteration in bacterial viability upon direct freezing to −70 °C without cryoprotectant [70]. Ultimately, regardless of the use of buffer or not, direct stool freezing is a common crucial step in fecal material processing to conserve fecal microbiota composition, viability and metabolic functionality. Given that the buffer requirements may vary with the subsequent type of analysis and certain preservative buffers might cause divergence compared to samples directly stored to −80 °C [76], it would be best to store the samples without buffer. Upon defreezing, the buffer of choice for subsequent steps can be added by each respective researcher, depending on the nature of their study.

### 2.6. Temperature and Time for Processing and Storage of Fecal Material

When considering the optimal temperature and time for processing and storage of fecal material, it is difficult to pick the exact combination because each study performs their stability test in a different manner. Despite the variability of storage conditions, available data support that colder storage temperature and shorter storage times are better for conserving microbial stability. Yet, in the case of manual fecal collection and sampling, “as cold and quickly as possible” is often not “as cold and quickly as optimal” due to logistical constrains. Storage time particularly plays an important role in intra-individual feces variation studies, where the stability of fecal microbiota of an individual is studied over time [77]. Table 1 shows available studies with temperature ranges from −80 °C to RT and time ranges from 15 min to 5 years for processing and storage. From this overview, it follows that fecal samples should be processed at either RT for at most 4 h or at 4 °C for no more than 24 h. Later, the aliquots should be stored at either −20 °C for up to a few months or −80 °C for longer periods—up to 2 years.

### 2.7. Parameters to Evaluate Stability upon Storage with Respect to Composition, Viability and Metabolic Functionality

The correlation between composition and metabolic functionality of gut microbiota with host health and disease are widely studied (Appendix A). An additional parameter relevant for characterizing the suitability of fecal sampling and storage conditions would be the viability of the isolated fecal bacteria. This especially holds for studies where one wants to isolate specific strains and/or investigate the origin of a specific metabolic reaction or pathway. Appendix A shows correlation of gut microbiota at phylum, family, genus and species level, which are the levels mostly studied for characterizing microbiota composition. Fecal microbiota composition is often performed with 16 s rRNA down to the genus level [81], yet this is shown to not be sufficiently deep in some studies where differences of specific microbiota occur at species or even sub-species level, requiring a deeper sequencing technique such as metagenomics [82]. Ideally, composition, viability and metabolic functionality of fecal microbiota should be tested because, when the bacteria are present, they might not always be viable and/or have their full metabolic capacity [11]. To test viability, one may use flow cytometry, to discriminate live and dead cells, and/or stool culturing on different media and under various conditions, to quantify the number of cultivable species [83]. Appendix A presents an overview of the many reactions that gut microbiota facilitate, which can be characterized by the catalytic efficiency measured with selected model compounds [84] or through metabolomics [85]. As quality control for the temporal stability [86] of gut microbiota over processing and storage conditions, composition, viability and metabolic functionality should be evaluated to avoid conclusions that can be mistaken for significant effects of a study. These studies will also facilitate a better characterization of the interindividual variability in composition, viability and metabolic functionality that may otherwise also act as a confounder in subsequent studies.

## 3. Conclusions and Future Perspectives

Studies on the gut microbiome protocols report a wide variety in the methods for human fecal sample collection, processing and storage. Only a limited number of studies exist on the actual stability of the fecal gut microbiota under different collection, processing or storage conditions in terms of either the microbial composition, viability or metabolic functionality. The aim of the present mini-review was to present an overview of the currently available data on collection and storage of fecal samples and consequences for stability of the samples, in order to define the requirements for an automated protocol to collect and store human fecal samples for research purposes in order to facilitate the process from toilet to freezer.

At the current state of the art, some tools for human fecal sample collection and processing are already available, such as fecal collection containers with a spoon [87], where processing is later performed in an anaerobic chamber [69]. Although this method is not standardized and is low-throughput, various research laboratories continue using this manual procedure in collecting human fecal materials because of different reasons such as financial (cheaper than machine at current status), time (relatively faster than learning how to use a new machine) and reproducibility (major change in current method decreases comparability with previously obtained results).

An automated protocol is, however, preferable because of the following considerations: (1) Post-defecation conditions are never 100% anaerobic, allowing exposure to air. For an automatic smart toilet combined with vacuum flush [88], this exposure-to-air period may be significantly reduced compared to manual collection by participants without the use of an anaerobic liquid or system [22]; (2) Fecal material processing often takes longer than 1 h after defecation to freezer due to transportation, homogenization and aliquoting [21,22], making it too laborious for a high-throughput study objective. Moreover, afterwards, aliquots need to be thawed, mixed with buffer of choice, centrifuged and filtered before analysis, adding to the extra processing time.

Based on the requirements above and Table 1, a plausible design for an automated fecal collection process is illustrated in Figure 1 and includes the following: (1) toilet with urine and feces separation; (2) once stool is excreted at RT within 10–15 min, stool should be vacuum flushed to the anaerobic chamber [22,54,78,79]; (3) within the anaerobic chamber, temperature should be set to 4 °C when fecal samples are homogenized and aliquoted within ≤1 h [75,76,89,90]; (4) still within the anaerobic chamber, aliquots are transferred to built-in freezer in the same device; (5) transfer of frozen samples to freezer within 10–15 min; (6) usage of −20 °C and −80 °C freezers for short term (2–3 months) and long-term (2 years) storage, respectively; (7) composition should be characterized by at least 16 S RNA analysis but, preferably, by deeper sequencing techniques, viability by flow cytometry or stool culturing in different conditions and metabolic functionality by an as-yet undefined but generally to be accepted set of fingerprinting metabolic conversions, reflecting the major types of biochemical conversions know to be catalyzed by the intestinal microbiota.

Although every study is different and a ‘one-size-fits-all’ protocol cannot exist [21], an automated protocol for the step of human fecal collection can increase the efficiency and allow gut microbiota studies to recruit more participants. The automated protocol/technology should be compact and portable so that it can be installed in research facilities but also be brought home when participants cannot come to the research facilities. Altogether, the overview of current practices for collection, processing and storage of fecal samples for studies on the composition, viability and metabolic functionality of the human intestinal microbiota, provides guidance of requirements for an automated protocol/technology to ensure fecal microbiota stability across processing and storage conditions.

## Figures and Tables

**Figure 1 biomedicines-11-02658-f001:**
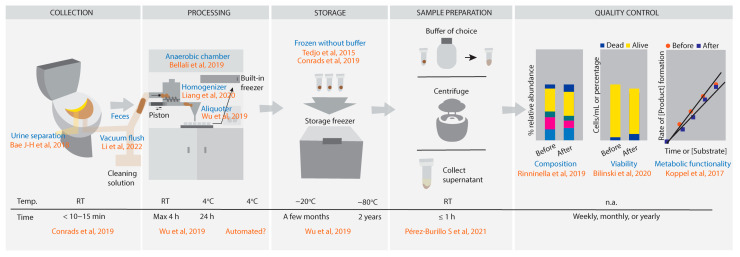
Automated protocol of fecal sample collection from toilet to freezer: collection, processing, storage and quality control [1,21,22,47,55,58,67,74,75,83,91]. The colors in bacterial composition, quality control are meant for illustrative purpose. Colours are only for illustration of different bacterial composition.

**Table 1 biomedicines-11-02658-t001:** Overview of processing and storage temperature and time across different studies. The coloring indicates whether the conditions indicated resulted in stable samples (green shading) or in samples that showed variability due to sample instability or deterioration (red shading). When this information was absent, no shading is included.

Processing			
Temperature	Time	Remarks	Reference
RT	3 h	After defecation. Stability not indicated.	[62]
20 ± 2 °C	<24 h	Non-significant effects at <24 h, but deletrious effects above this temperature.	[69]
>20 °C	>24 h	Samples should not be exposed to >20 °C if time from transformation to transplants exceeds 24 h.	
37 °C	>24 h	Extinction and proliferation of genus specific to each individual. Metabolomic fingerprints deviates even more as time increases.	
RT	4 h	After stool sample collection, it is acceptable for samples to reach the lab in this condition without preservation [54,78,79].	[22]
4 °C	24 to 48 h	Keep the sample in cold storage when transported from participant to laboratory staff handling.	
RT (with stabilizer)	Up to 4 weeks	Stabilizers such as RNALater, 95% ethanol, Omnigene-Gut (DNA Genotek, Stittsville, Ottawa, ON, Canada) and FTA card (Sigma-Aldrich, St. Louis, MO, USA) can protect stool DNA from degradation [80].	
4 °C (without stabilizer)	Up to a week	Fecal materials are processed without stabilizer [80].	
RT	Short-term	Short-term RT storage showed minimal effect on microbiome and metabolome profiles.	[55]
Storage			
Temperature	Time	Remarks	Reference
−20 °C	Until further use	Storage condition. Stability not indicated.	[62]
4 °C	Short-term	Ideal for short shipment.	[69]
−80 °C then rapid thaw at 37 °C	3 months	Stable fecal sample transplants in maltodextrin–trehalose solutions within this observation period.	
RT	Greater than 15 min	Storage with or without buffer showed considerable divergence compared to −80 °C samples.	[74]
4 °C	24 h	Does not alter microbiota compared to −80 °C samples.	
−20 °C	72 h	Does not alter microbiota compared to −80 °C samples.	
−80 °C	n.a.	Freezing significantly decreases overall viability to around 25% but does not significantly change viable microbiota composition compared to fresh anaerobically processed specimen.	[50]
−80 °C (with stabilizer)	5 years	Long term stool storage with RNAlater had only limited effects on human fecal microbiota composition.	[71]
−20 to −80 °C	As soon as possible	Once samples reach the laboratory, they should be stored in this condition. Stability not indicated.	[22]
−20 °C	A few months	Stable composition in microbial community.	
−80 °C	2 years to long-term storage	Stable composition in microbial community.	

## Data Availability

Appendix A is available.

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
