# Peer review of "From-Toilet-to-Freezer: A Review on Requirements for an Automatic Protocol to Collect and Store Human Fecal Samples for Research Purposes"

_biomedicines, 2023, doi:10.3390/biomedicines11102658_

Round 1
Reviewer 1 Report
Increasing evidence shows the crucial role of gut microbiota in human health and understanding the microbial composition difference and developing culturomics methods became hot topics recently. However, the very first question is about how gut microbiota is collected from donor, i.e., via stool, and what the difference between multiple collection modalities. Widjaja, Frances et al. summarized the current protocols used for human fecal samples collections and described the potential issues in different steps. Overall the manuscript is well-written and well-organized and the authors pointed out a few very important but currently under-characterized issues which will be a critical alert to other researches in this field. Thus, I will support the publication of this work if the following issues could be addressed by the authors.
In general, the authors need to describe the “performance” of fecal sample collection in two different ways: (1) to characterize compositional profile via DNA (2) to obtain living bacteria for culturing or colonizing. For example, while some collection methods will protect microbial DNA better (like RNAlater or other non-lytic or lytic buffer), it will kill bacteria thus greatly decrease the viability of living cells for culturing and colonzing studies. Currently this concept is kind of mixed in the manuscript and causes confusing (such as table-1 and other related paragraghs).
Supplementary material is missing in reviewing material. Moreover, I would suggest the author to have a simple figure to demonstrate the different aspects of gut microbiota rather than supplementary tables (S1 and S2).
Line-65: good summary of the donor inclusion criteria.
Line-92 to 111: authors should summarize what is the current widely used feces collection/holding kit, for example, some commercially available kits like Zymo feces catcher or Fisherbrand Commode Specimen Collection System.
Line-120: I think it is a common sense that freeze-thawing cycles would kill/lyse bacteria, so the viability will be greatly impaired. The authors need to discuss more on this. Maybe the overall composition of viable bacteria won’t change too much?
Line-124 to 149: good job on summarizing this. I agree this is a very important part but actually largely missing in the current studies.
Line-184: see comments for Line-120, this sentence is actually conflict with what is mentioned before so need to articulate more.
Line-18: Typo 4oC
Line-215: “Error! Reference source not found”
Author Response
Please see the attachment Section 1 Reviewer 1.

Reviewer 2 Report
Overall, this is a well written review publication and offers a good overview summarizing the state of knowledge concerning fecal sampling, processing and storing as a source of human intestinal microbiota. This manuscript shows rich content, providing a deep insight for some works: the study is within the journal’s scope, and I found it to be well-written, providing sufficient information. However, before publication some points need to be clarified.
My comments:
Line 17 – please explain what RT stands for
Line 18 – change to “oC”
Line 53 – please present a goal of this review and add a short methodology of it.
Line 127, 133, 134, etc. – please avoid presenting references as “Santiago et al. (2014)”. Use numbers instead.
Line 215 – please check reference as “Error! Reference source not found” appeared.
Table 1 – please delete 1st column. In the last column (references) all details concerning authors can be found.
Line 245 – I do not understand the idea of “Discussion” in review article. To whom and what the authors want to discuss? The review article is per se a form of discussion.
Line 299 – please add some conclusion and future perspectives.
Author Response
Please see the attachment Section 2 Reviewer 2

Reviewer 3 Report
This authors in this article discusses the challenges and considerations in microbiota studies that rely on fecal samples. It proposes the development of an automated protocol for sampling, processing, and storing these samples, addressing factors such as donor criteria, smart toilet technology for urine-feces separation, homogenization, aliquoting, choice of buffer, temperature, time limits, and quality control. The authors highlight the need for standardization and increased efficiency in fecal sample collection procedures, ultimately recommending specific conditions for sample processing and storage to improve microbiota research. Quality control protocols should assess both microbial composition and functionality in these samples.
Upon my thorough review of the paper, I must express my intrigue with its genuinely interesting and valuable findings. The authors have made a systematic effort to address the topic and have consolidated valuable and useful information in one place. I believe that their work unquestionably contributes to the field of science and is engaging to read. Everything is presented very systematically and clearly.
However, it appears that there may be a misunderstanding, as I am unable to locate or access the supplementary materials referenced in the text. While the written content is compelling, I regret to inform you that I have not had the opportunity to review the additional materials due to their inaccessibility. It is crucial that the mentioned materials be made available to reviewers, as they are integral to the comprehensive evaluation of the work. Once the supplementary materials become accessible, we can reconsider the possibility of accepting the paper. The manuscript is well-written and highly engaging, but the absence of the referenced supplementary materials prevents me from recommending it for publication at this time.
I included few specific comments to be addressed:
1. Line 109-111: I believe it would be truly wonderful to have such a device for human fecal excreta collection, but since it is not possible in many parts of the world, I would recommend revising this sentence to suggest, if feasible. The term 'prerequisite' may be accurate, but it might be an impossible mission in some places where research is undoubtedly taking place. Therefore, I would recommend a minor revision to this sentence
2. Line 215: Please correct the highlighted error, which suggests that a reference source is missing.
3. Line 272: Please correct the highlighted error, which suggests that a reference source is missing.
4. Lines 273-274: Please correct the highlighted error, which suggests that a reference source is missing.
Author Response
Please see the attachment Section 3 Reviewer 3
